# Factors Associated with Pro-Environmental Behaviors in Israel: A Comparison between Participants with and without a Chronic Disease

**DOI:** 10.3390/ijerph192013123

**Published:** 2022-10-12

**Authors:** Shiri Shinan-Altman, Yaira Hamama-Raz

**Affiliations:** 1School of Social Work, Bar-Ilan University, Ramat Gan 5290002, Israel; 2School of Social Work, Ariel University, Ariel 40700, Israel

**Keywords:** chronic disease, climate change exposure, climate change risk appraisal, environmental self-efficacy, collective efficacy, pro-environmental behaviors

## Abstract

This study examined differences regarding climate change pro-environmental behaviors (PEBs), comparing between individuals with chronic diseases and those without. A cross-sectional survey was conducted among 402 adults, of whom 25% had a chronic disease. Participants completed measures for PEBs, climate change exposure (i.e., exposure to its effects), climate change risk appraisal, environmental self-efficacy, collective efficacy, and sociodemographic variables. Results revealed a significant difference between participants with and without chronic diseases in climate change risk appraisal. Having a chronic disease was associated with higher climate change risk appraisal (β = 0.16, *p* < 0.001), which in turn was associated with higher collective efficacy (β = 0.29, *p* < 0.001). The latter was associated with more PEBs (β = 0.10, *p* = 0.049). Furthermore, higher climate change exposure was associated with higher climate change risk appraisal (β = 0.49, *p* < 0.001), which in turn was associated with collective efficacy (β = 0.29, *p* < 0.001). The latter was associated with more PEBs (β = 0.10, *p* = 0.049). In addition, higher climate change exposure was directly associated with both self-efficacy (β = 0.33, *p* < 0.001) and collective efficacy (β = 0.10, *p* = 0.049), which in turn were associated with more PEBs (β = 0.28, *p* < 0.001 and β = 0.10, *p* = 0.049, respectively). This study highlights the need to provide efficacy-enhancing information in climate change messaging for PEBs in general. A threat component in environment-relevant messages for people with chronic diseases, specifically, should also be adopted.

## 1. Introduction

Climate change is one of the most serious threats to both physical and mental health as its impacts and risks are becoming increasingly complex and more difficult to manage [1]. As a result of climate change, human health, including mental health, may be affected directly through extreme heat, cold, drought, or storms, or indirectly via changes in air quality, water availability, food provision and quality, and other stressors [2,3,4].

In Israel, where the present study was conducted, summers are hot and winters are mild, with significant changes in rainfall patterns from year to year [5]. Impacts of climate change in Israel include increasing temperatures, which could lead to drier conditions and stronger storms, as well as lower rainfall [6]. Further health risks derived from climate change include air pollution, vector-borne diseases, and heat stress (i.e., heat-related illnesses). Sources of ambient air pollution in Israel include anthropogenic activity (such as energy production, industry, transportation, and infrastructure) and also natural phenomena (such as desert dust) [6].

Given the possible climate change-related risks to mental and physical health [7], people with existing medical conditions such as cardiovascular disease, diabetes, obesity, and respiratory diseases are more likely to suffer. Their illness conditions may worsen, and increased exposure to extreme heat, extreme weather events, and poor air quality may even lead to death [8]. The severity of these risks will depend on the ability of public health systems to address or prepare for these changing threats [9]. A systematic review of the impacts of cyclone, flood, and storm-related disasters on health care for people with noncommunicable diseases found that people with cancer, diabetes, and cardiovascular diseases sustained an increased risk of exacerbated health problems following such disasters. These outcomes were due to a range of factors including disruption of transport, weakened health systems including drug supply chains, loss of power, and population evacuations [10]. In this context, a recent study conducted in Israel revealed that living in the Haifa Bay Area was associated with a higher adjusted risk of asthma in adolescents aged 17 compared with their same-aged counterparts in the non-Haifa Bay Area [11]. Likewise, Greenberg and colleagues [12] found in their study among young Israeli males that exposure to industrial- and traffic-related air pollution had a detrimental effect on their respiratory health and that such exposure was positively associated with asthma prevalence. Thus, as the health impacts of climate change will be determined mainly by the vulnerability of populations, their resilience to the current climate change rate, and the extent and pace of their adaptation [9], it seems important to focus on vulnerable populations such as those with chronic diseases.

Individuals’ pro-environmental behaviors (PEBs) are crucial in the climate change challenge as they are the drivers of larger processes of change (e.g., public pressure) and as policy decisions are taken by individuals [13]. The commonly accepted definition for PEBs is purposeful action that can reduce a negative impact on the environment [14]. Pro-environmental behaviors include different kinds of operationalized behaviors which can be summed up as three major environmental behaviors: waste reduction, reuse, and recycling [15]. Although the vast majority of people agree that protecting the environment is important [16], only a relatively small percentage of Europeans have actually practiced PEBs (e.g., cutting down their energy consumption or avoiding single-use plastic goods) [17]. As such, it seems even more necessary to explore whether there are differences in PEBs between those who have chronic diseases (i.e., climate change may exacerbate their diseases) and those who do not have chronic diseases.

Different psychological and sociological models have been developed to explain the complexity of the determinants underlying PEBs. However, Li and colleagues [15] noted “what’s shaping PEB is so complicated that a single model cannot encompass all the relevant factors” (p. 29). In line with this notion, we adopted several components from the transactional model of stress and coping [18] as possible predictors of PEBs. Specifically, according to Lazarus and Folkman [18], individuals are constantly appraising and updating their beliefs concerning stimuli (e.g., climate change) within their environment. When people appraise stimuli as threatening, challenging, or harmful (i.e., primary appraisal), they can then evaluate their ability to cope with the consequences of their interactions with the environment (i.e., secondary appraisal—environmental self-efficacy and collective efficacy). Prior experience or being exposed to similar situations previously provides a frame of reference to determine the options available for dealing with the situation.

In accordance with the transactional model of stress and coping [18], the following predictors of PEBs among people with chronic diseases vs. people without chronic diseases were examined: climate change exposure, climate change risk appraisal, environmental self-efficacy, and collective environment-related efficacy (hereafter collective efficacy) (See Figure 1).

*Climate change exposure* relates to the experience one has had with climate change effects. Ogunbode and colleagues [19] claimed that “proximizing climate change (i.e., framing the issue as more immediate, relevant, and real) has the potential to motivate individuals to act pro-environmentally by (1) making climate change consequences more personally relevant and easier to visualize, (2) creating feelings of personal vulnerability and concern, and (3) decreasing the psychological distance of climate change among individuals with a responsibility or capacity for action” (p. 446). According to McDonald and colleagues [20], people often draw on their personal experiences with extreme weather to make inferences about the reality of climate change. In addition, recent research conducted in China revealed that exposure to climate change-related information on social networking sites had a direct positive effect on users’ PEBs [21]. Likewise, perceived climate change exposure has been associated with behavior aimed at reducing one’s carbon footprint, such as recycling [22].

Climate change risk appraisal refers to people’s monitoring of events with regard to their well-being [23]. Personal well-being can be threatened if a situation is appraised as impairing or threatening personal goals, health, or identity [24]. In previous research it was found that climate change risk appraisal had a positive and significant influence on attitudes toward climate change, which in turn had a positive and significant influence on PEBs [25]. In this regard, participants with asthma and chronic obstructive pulmonary disease strongly perceive environmental factors such as ambient outdoor exposure, including pollen and air pollution, to impact their symptoms. Therefore, they seek out environmental information daily through a variety of sources, with a preference for mobile apps and TV [26].

Environmental self-efficacy is defined as “people’s beliefs in their capabilities to produce desired effects in the environment by their own actions” [27] (p. 6). This self-belief factor may influence individuals’ responsible environmental behavior and attitudes, in that individuals who believe they have the requisite skills to do so are more likely to manifest more responsible environmental behavior [28]. In this context, Díaz and colleagues [29] found that environmental self-efficacy was an important predictor of PEBs in a developing country’s population. Similarly, Huang [30] revealed that environmental self-efficacy positively predicted proactive environmental behavior.

Collective efficacy is a group’s belief in the group’s coping abilities to reach desired goals [24,31]: all individuals are influenced by their own efficacy beliefs regarding the goal at hand, as well as their beliefs regarding the abilities of others involved in the same movement, and, finally, regarding the movement as a whole [32]. It has been suggested in previous studies that collective efficacy beliefs are a stronger predictor of PEBs than are self-efficacy beliefs [24,33], although in these previous studies the constructs have not always been assessed simultaneously [24]. Moreover, Jugert and colleagues [34] found that increased perceived collective efficacy, followed by increased self-efficacy, mediated the positive association between collective efficacy manipulation and pro-environmental intentions among adult participants in Germany and Australia.

In sum, based on the main assumptions of the transactional model of stress and coping [18], we employed a novel research model with the following two aims: 1. To examine possible differences regarding climate change PEBs via making a comparison between those with chronic diseases and those without and, as such, to test the proposed model, and 2. To examine the associations between climate change exposure, climate change risk appraisal, environmental self-efficacy, collective efficacy, and PEBs. The latter aim involved an evaluation of whether climate change risk appraisal, environmental self-efficacy, and collective efficacy mediated the link between climate change exposure and PEBs (indirect effect). As climate change constitutes a significant threat to public health, and people who have chronic diseases are particularly vulnerable to severe health consequences, these aims are critical.

## 2. Material and Methods

### 2.1. Hypotheses

Our hypotheses were as follows:

**Hypothesis** **1** **(H1).**
*Participants with a chronic disease (compared to those without) would report higher climate change exposure, higher climate change risk appraisal, stronger environmental self-efficacy and collective efficacy, as well as higher PEBs.*


**Hypothesis** **2** **(H2).**
*Climate change exposure, climate change risk appraisal, environmental self-efficacy, and collective efficacy would be positively associated with PEBs.*


**Hypothesis** **3** **(H3).**
*Climate change risk appraisal, environmental self-efficacy, and collective efficacy would mediate the relationship between coping with a chronic disease and PEBs.*


### 2.2. Procedure

In this study we used an internet panel of Israelis that adheres to Israel’s Bureau of Statistics in key sociodemographic factors including age, gender, and marital status, and represents the general population [35]. Participant eligibility included being 18 years of age or older and being fluent in Hebrew. The study was approved by the first author’s university’s institutional review board (IRB; approval No. 052202). Data collection was conducted during the first two weeks of June 2022, when the COVID-19 outbreak was under control and normal routines were in place. Each participant signed an electronic informed consent form before accessing the questionnaire.

### 2.3. Measures

The following validated self-report measures were used: (the questionnaire used for this study is available as an online supplementary material):

Pro-environmental behaviors (PEBs) were assessed via Homburg and Stolberg’s scale [24]. Two PEB types were assessed: Six items measured private-sphere environmentalism, in particular regarding mobility, energy and water consumption (e.g., “When possible, I use public transport instead of going by car”). Two items measured non-activist public-sphere behavior, in particular social commitment, which refers to participation in environmental protection actions (e.g., “I participate in protest campaigns or demonstrations for environmental protection”). Participants were asked to rate the extent to which they agreed or disagreed with each statement on a 5-point Likert scale ranging from 1 = strongly agree to 5 = strongly disagree. A mean score was calculated; a higher score indicated higher PEB levels.

Climate change exposure was assessed via Clayton and Karazsia’s subscale of a climate change anxiety measure [36]. This subscale consists of the three following statements: (1) “I have been directly affected by climate change”; (2) “I know someone who has been directly affected by climate change”; (3) “I have noticed a change in a place that is important to me due to climate change” [36] (p. 4). Participants were asked to rate the frequency of each of the three statements on a 5-point Likert scale ranging from 1 = never to 5 = often. A mean score was calculated; a higher score indicated higher levels of climate change exposure.

Climate change risk appraisal was assessed via Homburg and Stolberg’s scale [24]. Two types of risk appraisal were assessed (threat and harm). Three items measured threat: (1) “I am not worried about the health consequences of pollution”; (2) “I feel that my health is threatened by pollution in everyday life”; (3) “The thought of this pollution makes me uneasy.” An additional three items measured harm: (1) “So far, pollution in everyday life has not harmed me”; (2) “My health has become worse because of the pollution in everyday life”; (3) “I have lost hope, because pollution has just gotten worse and worse” [22] (p. 12). Participants were asked to rate the extent to which they agreed or disagreed with each statement on a 5-point Likert scale ranging from 1 = strongly agree to 5 = strongly disagree. A mean score was calculated; a higher score indicated higher levels of climate change risk appraisal.

Environmental self-efficacy was assessed via Homburg and Stolberg’s scale [24]. Participants were asked to rate the extent to which they agreed or disagreed with nine statements on a 5-point Likert scale ranging from 1 = strongly agree to 5 = strongly disagree (e.g., “I know how to take precautions against pollution in everyday life”; “When I hear about pollution of this kind, I usually have various ideas of how to deal with it”). A mean score was calculated; a higher score indicated higher levels of self-efficacy.

Collective efficacy was assessed via Homburg and Stolberg’s scale [24]. Participants were asked to rate the extent to which they agreed or disagreed with six statements on a 5-point Likert scale ranging from 1 = strongly agree to 5 = strongly disagree (e.g., “I am confident that together we can solve the problem of pollution”; “I am convinced that we as humans can together create a healthier environment, even if the opportunities to do so should decrease”). A mean score was calculated; a higher score indicated higher levels of collective efficacy.

Sociodemographic variables included gender, age, years of education, marital status (married/divorced/widowed/single), number of children, employment status (full-time/part-time, salaried/freelance, employed/unemployed/retired/stay-at-home-parent), religiosity (secular/traditional-not so religious/traditional-religious/Orthodox).

Health information included smoking (yes/no), and having a chronic disease (yes/no, and if “yes” indicating which disease). Self-rated health was assessed via a single question: “In general, how do you rate your health?” The scale ranged from 1 to 4 (1 = bad to 4 = excellent) [37].

### 2.4. Statistical Analyses

Data were analyzed using SPSS ver. 28. As the internet panel consisted of about 130,000 Israelis, we calculated the sample size of a random sample drawn from it. Following the criteria of a 95% confidence level, and a 5% sampling error, the required sample size would be 384 participants. The sample size was further calculated with G*Power 3 [38,39], showing that for a multiple regression analysis with up to 15 predictors, a moderate-low effect size f^2^ = 0.10, α = 0.05, and power = 0.99, the required sample size would be *N* = 379 participants. Descriptive statistics were used to describe the participants’ sociodemographic variables and the research variables. Cronbach’s α was used for internal consistency. Pearson correlations were calculated to assess the associations between the research variables. The strength of correlation was as follows: 0–0.20, weak; 0.21–0.50, moderate; 0.51–0.80, good; and 0.81–1-00, excellent. Differences in the study variables, between participants with and without a chronic disease, were calculated with a series of *t*-tests. The relationships between the study variables and the sociodemographic variables were examined with Pearson correlations (r), Cramer’s V for categorical variables, and *t*-tests. A multiple linear hierarchical regression was calculated for PEBs with the study variables. Age, years of education, and having a chronic disease were entered in the first step, and the study variables in the second. The research model was examined with path analysis using AMOS software, ver.28. Continuous variables were standardized. Measures of model fit (within the AMOS software)—namely, chi-square, NFI (normed fit index), NNFI (non-normed fit index), CFI (comparative fit index), and RMSEA (root mean square error of approximation)—were used as measures of fit. Age, years of education, and having a chronic disease were entered as covariates. Climate change exposure was defined as the independent variable; climate change risk appraisal and environmental self- and collective efficacy were defined as the mediating variables; and behavior (PEBs) was defined as the dependent variable. Correlations among the covariates were included in the model, as well as between the two mediators of environmental self-efficacy and collective efficacy. Mediation was examined as part of the path analysis, with bootstrapping of 5000 samples and a bias-corrected 95% confidence interval. Bootstrapping was used to generate an empirically derived representation of the sampling distribution of the indirect effect, and to allow us to derive estimated standard errors and confidence intervals [40].

## 3. Results

### 3.1. Participants

Out of all 130,000 panelists, 5378 panelists were invited to participate in the online survey. To create a representative sample of the population, once quotas by gender, age, and education were reached for each parameter, the survey was closed. Out of the 5378 panelists, 402 (7.5%) people completed the study’s questionnaire and participated in the study. As can be observed in Table 1, the sample included 101 participants with a chronic disease—specifically, asthma (*n* = 21); cancer (*n* = 10); high blood pressure (19); cardiac disease (*n* = 8); allergy (*n* = 25); diabetes (*n* = 14); skin (*n* = 4)—and 301 participants without a chronic disease. Among participants with a chronic disease, 45.5% were male and 54.5% were female, and among participants without a chronic disease, 49.5% were male and 50.5% were female. The mean age among participants with a chronic disease was significantly higher than among participants without. In addition, a significant difference was found between the two groups in employment status. Perceived health status was significantly higher (better) among participants without a chronic disease. There were no significant differences between the groups in mean number of years of education, marital status, mean number of children, religiosity, or smoking.

### 3.2. Correlations between Study Variables

Table 2 summarizes the means, SDs, ranges, Cronbach’s α, and correlations for the study’s variables. Significant positive and weak-to-moderate correlations were found among the study variables. Higher climate change exposure, higher climate change risk appraisal, higher environmental self-efficacy, higher collective efficacy, and more PEBs were all interrelated.

### 3.3. Differences in Study Variables Comparing People Coping with Chronic Diseases to People without

Participants with a chronic disease appraised the risk of climate change more highly than participants without. This difference had a small effect size, and other differences were not significant (see Table 3).

### 3.4. Inter-Relations among Study Variables and PEB Predictors

The sociodemographic and health-related variables were unrelated to PEBs (*p* = 0.109 to *p* = 0.926). A few weak relationships were found between the sociodemographic variables and the other study variables. Specifically, participants’ age was positively associated with environmental self-efficacy (r = 0.12, *p* = 0.015), and years of education was positively associated with both environmental self-efficacy (r = 0.11, *p* = 0.028) and collective efficacy (r = 0.14, *p* = 0.007). Having a chronic disease was associated with higher climate change risk appraisal (Table 3), as well as with participants’ self-rated health (Cramer’s V = 0.34, *p* <.001). Self-rated health was negatively associated with climate change risk appraisal (r = −0.18, *p* < 0.001). Furthermore, participants who smoked (or used to) had lower environmental self-efficacy (Mean = 2.50, SD = 0.74) than participants who did not smoke (Mean = 2.68, SD = 0.72) (*t*_(400)_ = 2.37, *p* = 0.018, *d* = 0.25). In line with the aforementioned, our previously outlined hypotheses were examined while including age, years of education, and having a chronic disease in the equations.

Table 4 shows that the regression model was significant, with 17% of the variance in PEBs being explained in it. Higher climate change exposure, higher environmental self-efficacy, and higher collective efficacy were associated with more PEBs.

### 3.5. Research Model Assessment

The research model was examined with a path analysis (AMOS ver.28), as shown in Figure 2. Age, years of education, and chronic disease (1-yes, 0-no) were entered as covariates. The study model was found to fit the data well: χ^2^(13) = 4.37, *p* = 0.348, *NFI* = 0.965, *NNFI* = 0.992, *CFI* = 0.996, *RMSEA* = 0.016. As shown in Figure 2, having a chronic disease was directly related with climate change risk appraisal, such that participants who had a chronic disease had a higher risk appraisal in regard to the effects of climate change than participants who did not. Furthermore, climate change exposure was positively associated with climate change risk appraisal, environmental self-efficacy, and collective efficacy, and with PEBs. Climate change risk appraisal was positively associated with collective efficacy, and both environmental self-efficacy and collective efficacy were positively associated with PEBs.

In addition, two main indirect relationships were found to be significant. The first involved chronic disease as the independent variable and PEBs as the dependent variable (indirect effect = 0.005, 95% CI = 0.001, 0.015, SE = 0.003, *p*= 0.011). As shown in Figure 2, having a chronic disease was associated with a higher climate change risk appraisal, which in turn was related to higher collective efficacy. The latter was then related to more PEBs.

The second significant indirect relationship involved climate change exposure as the independent variable and PEBs as the dependent variable (indirect effect = 0.139, 95% CI = 0.071, 0.210, SE = 0.035, *p* < 0.001). The results in Figure 2 reveal that higher climate change exposure was associated with both higher environmental self-efficacy and collective efficacy, which in turn were associated with more PEBs. In addition, higher climate change exposure was associated with higher climate change risk appraisal, which in turn was associated with higher collective efficacy. The latter was then associated with more PEBs.

## 4. Discussion

Climate change constitutes a major threat to public health, and people who have chronic diseases are particularly vulnerable to severe health consequences [8]. In the present study, we sought to investigate possible differences regarding climate change PEBs, making a comparison between participants with and without a chronic disease. In addition, we examined associations between climate change exposure, climate change risk appraisal, environmental self-efficacy, collective efficacy, and PEBs. The results revealed a significant difference between participants with and without chronic diseases in climate change risk appraisal. Having a chronic disease was associated with higher climate change risk appraisal, which in turn was associated with higher collective efficacy. The latter was associated with more PEBs. Furthermore, higher climate change exposure was associated with higher climate change risk appraisal, which in turn was associated with collective efficacy. The latter was associated with more PEBs. In addition, higher climate change exposure was directly associated with both self-efficacy and collective efficacy, which in turn were associated with more PEBs.

In line with previous literature [24], we found that participants with a chronic disease appraised the effects of climate change as more negative than participants without one. In other words, compared to participants without a chronic disease, participants with a chronic disease perceived the effects of climate change as having more severe consequences. This finding suggests that participants with a chronic disease may feel threatened because they have appraised climate change as impairing or threatening their personal goals, health, or identity [24]. Indeed, chronic diseases can make individuals more sensitive to climate change exposure, given the increasing potential for health impacts and worsening symptoms [8].

In this regard, the U.S. Global Change Research Program (USGCRP) Climate and Health Assessment noted that people with chronic diseases often need consistent medication or medical care. Extreme weather events can disrupt care due to evacuations, transportation system or health infrastructure damages, or power outages. Additionally, some medications for chronic conditions can affect the body’s response to heat, putting people at greater risk for heat illnesses as the climate warms. Moreover, individuals with certain medical conditions can have compromised immune systems, potentially making them more prone to extreme heat-related reactions, insect- and tick-related diseases, and water-related illnesses [9].

However, in all of the other study’s variables (climate change exposure, environmental self-efficacy, collective efficacy, PEBs), there were no significant differences between the two groups. This area, we believe, is still one that requires further research given that the study of primary and secondary appraisals of patients with chronic diseases in the context of climate change has not been sufficiently developed as yet [41].

Our results indicated that higher climate change exposure, higher environmental self-efficacy, and higher collective efficacy were associated with more PEBs. These associations are in line with the transactional model of stress and coping [18] and with previous studies examining PEBs [42]. Self-efficacy and collective efficacy may serve as buffers against the source of stress (e.g., climate change) and promote adaptive behavior such as PEBs. It should be noted that although climate change exposure, environmental self-efficacy, and collective efficacy have been used separately in previous literature to predict environmental behaviors [42], they have not been thoroughly tested in combination, although considerations of collective efficacy (i.e., are we as a group capable of dealing with this problem?) should play a prominent role in motivating individuals to engage in pro-environmental action [34]. Yet, the regression model in the current study was significant, with 17% of the variance in PEBs being explained in it, suggesting the need to include additional variables in future studies in an attempt to better explain the complexity of PEBs among people with and without chronic diseases.

Although climate change puts people with chronic diseases at higher risk than it does the general population [8], this unique population has only been examined in a handful of studies with respect to climate change perceptions [41]. In the present study, we have shed light on the importance of examining people with chronic diseases as we found that having a chronic disease was associated with a higher climate change risk appraisal, which in turn was related to higher collective efficacy. The latter was then related to more PEBs. This finding highlights the importance of examining the mechanism of PEB promotion by examining patients’ assessments of climate change as a threat and promoting collective efficacy to reduce this perceived threat.

Another finding of the present study relates to the mediating role of climate change risk appraisal, environmental self-efficacy, and collective efficacy in the relationship between climate change exposure and PEBs. This result confirms findings from a recent study [42] which revealed that climate change exposure had the potential to make people appraise climate change as threatening and to behave in a way that might reduce this threat. In a similar way, climate change exposure might evoke personal and collective beliefs such as self-efficacy and collective efficacy, which might encourage people to take protective actions, such as PEBs. In reference to this notion, in a previous study it was demonstrated that collective efficacy manipulations could increase pro-environmental intentions by increasing the perception that one’s group—and, through one’s group, oneself—is capable of effecting change [34].

In this study, overall, the PEBs mean score was moderate, relative to the scale range. This finding can perhaps be attributed to the Israeli context. In Israel, similar to the United States, recycling, for example, is either voluntary or adopted within environmentally conscious organizations [43]. The social context in Israel reflects the fact that PEBs are not always a default option. Pro-environmental behaviors require mostly voluntary motivation to contribute to solving environmental problems, as there are no specific rules requiring people to do so. In addition, the study was conducted early in the month of June, a time characterized by comfortable weather in Israel. If the study had been conducted during the winter or summer, the results may have been different as people are more likely to be worried about the reality and risks associated with climate change on hotter or colder days and months [44]. Another possible explanation can perhaps be found in a recent study which revealed that pro-environmental dispositions predicted PEBs only when the actors were prompted to experience a high rather than low sense of power (namely, feelings of control and autonomy) [17]. Thus, it might be that our participants reported a moderate level of PEBs as a reflection of their values and attitudes or a sense of power regarding climate change challenges.

The current study had several limitations. First, the use of a convenience sample does not allow us to generalize from the findings, nor does it provide an accurate representation of all people in Israel. Moreover, no information was gathered regarding the duration of the chronic diseases, medical therapies, or subjective suffering of those participants who reported having a chronic disease. Additionally, the response rate for completing the questionnaire was low, and characteristics of those refusing to participate were not assessed. Researchers in future studies should use probability samples to explore these important issues. Second, we used a cross-sectional design; as such, conclusions about directionality or causality in the relationships between the study’s variables should be arrived at cautiously. Third, the data refer only to the period in which they were collected. Given the dynamics of climate change, the rates may change over time. Further longitudinal research is needed in Israel and around the world. Finally, PEBs were self-reported, potentially leading to bias. Thus, caution should be applied in generalizing the results.

These limitations notwithstanding, the results of this study have important theoretical and policy-wise implications. Theoretically, although the transactional model of stress and coping [18] has been widely used as a framework for understanding coping with a health threat, it has rarely been used for understanding environmental behaviors. To the best of our knowledge, this study is the first to include these psychological adaptation variables (i.e., climate change exposure, climate change risk appraisal, environmental self-efficacy, and collective efficacy) and PEBs based on this model. As can be seen in the current study, however, this model can be particularly relevant to environmental behaviors because messages related to climate change often rely on using existential threats [45]. For policy-wise interventions, we suggest that the threat component can be an effective strategy for environment-relevant messages for people with chronic diseases. Thus, the media might use appropriate threat components in their messaging to increase perceived severity and motivate PEBs among people with chronic diseases. In addition, the results indicate that environmental self-efficacy and collective efficacy play a significant role in PEBs. This finding suggests that it is important to provide efficacy-enhancing information in climate change messaging for PEBs. For example, in climate change campaign messages, after communicating the idea that climate change is a serious problem that can threaten lives, the message should provide information that would increase people’s self-efficacy to engage in PEBs. Self and collective efficacy-enhancing information could include easy and specific guidelines for engaging in PEBs.

In conclusion, the current study makes a novel contribution to the literature in regard to PEBs among people who are coping with chronic diseases in comparison to those who are not. Climate change risk appraisal emerged as an important component that differed between the two groups and that was subsequently associated with collective efficacy and in turn with PEBs. Researchers in future studies should explore whether people with chronic diseases react or adapt differently to climate change threats.

## Figures and Tables

**Figure 1 ijerph-19-13123-f001:**
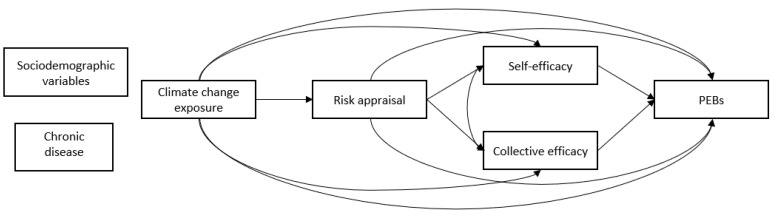
The study’s model.

**Figure 2 ijerph-19-13123-f002:**
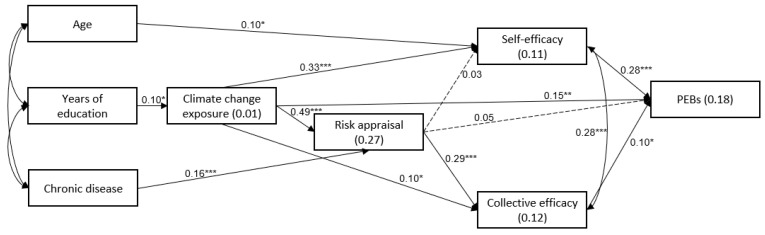
Path analysis for climate change exposure, climate change risk appraisal, environmental self-efficacy, collective efficacy, and pro-environmental behaviors (PEBs). * *p* < 0.05, ** *p* < 0.01, *** *p* < 0.001. Note: Solid lines: significant associations. Dashed lines: nonsignificant associations. Unidirectional arrows: path β values, bidirectional arrows- correlations. Values on arrows—β values; values within the rectangles—*R*^2^.

**Table 1 ijerph-19-13123-t001:** Participants’ sociodemographic variables.

	Participants with a Chronic Disease (*n* = 101)	Participants without a Chronic Disease (*n* = 301)	Comparison	*d*
Gender (% Female)	54.5	50.5	χ^2^ _(1)_ = 0.47, *p* = 0.491	0.07
Mean age (SD)	50.18 (15.31) ^a^	38.77 (13.61) ^b^	*t*_(156)_ = −6.66, *p* < 0.001	0.79
Mean number of years of education (SD)	14.17 (2.98) ^c^	13.59 (2.63) ^d^	*t*_(400)_ = −1.85, *p =* 0.065	0.21
Marital status (% Married)	55.4	54.2	χ^2^ _(1)_ = 0.05, *p =* 0.821	0.02
Mean number of children (SD)	1.92 (1.39) ^e^	1.69 (1.79) ^f^	*t*_(220)_ = −1.31, *p =* 0.191	0.14
Employment status (% Full time)	43.6	56.5	χ^2^ _(1)_ = 5.07, *p =* 0.024	0.23
Religiosity (% Secular)	53.5	51.8	χ^2^ _(1)_ = 0.08, *p =* 0.775	0.03
Smoking (% No)	62.4	63.5	χ^2^ _(1)_ = 0.04, *p =* 0.846	0.02
Perceived health status	2.80 (0.60)	3.24 (0.56)	*t*_(400)_ = 6.65, *p* < 0.001	0.76

SD = Standard Deviation; ^a^ Range: 19–70, ^b^ Range: 18–70, ^c^ Range: 9–24, ^d^ Range: 7–26, ^e^ Range: 0–6, ^f^ Range: 0–10.

**Table 2 ijerph-19-13123-t002:** Means, SDs, ranges, Cronbach’s α, and correlations for the study variables (*n* = 402).

Variables	1	2	3	4	5
1. Pro-environmental behaviors (PEBs)	-				
2. Climate change exposure	0.29 ***	-			
3. Climate change risk appraisal	0.20 ***	0.49 ***	-		
4. Environmental self-efficacy	0.37 ***	0.32 ***	0.14 **	-	
5. Collective efficacy	0.24 ***	0.24 ***	0.34 ***	0.33 ***	-
Mean	2.54	2.01	3.13	2.62	3.75
SD	0.60	0.95	0.71	0.73	0.84
Possible range	1–5	1–5	1–5	1–5	1–5
Actual range Cronbach’s α	0.60	0.86	0.73	0.89	0.93

SD = standard deviation; ** *p* < 0.01, *** *p* < 0.001.

**Table 3 ijerph-19-13123-t003:** Means, SDs, and *t* values for the study variables by chronic disease (*n* = 402).

	Participants with a Chronic Disease Mean (SD)	Participants without a Chronic Disease Mean (SD)	*t*_(400)_ (*p*)	*d*
Pro-environmental behaviors (PEBs)	2.53 (0.59)	2.55 (0.60)	−0.37 (*p* = 0.715)	0.03
Climate change exposure	2.03 (0.88)	2.01 (0.98)	0.18 (*p* = 0.857)	0.02
Climate change risk appraisal	3.33 (0.66)	3.06 (0.72)	3.42 (*p* < 0.001)	0.39
Environmental self-efficacy	2.62 (0.67)	2.61 (0.75)	0.10 (*p* = 0.924)	0.01
Collective efficacy	3.77 (0.65)	3.74 (0.90)	0.43 (*p* = 0.665)	0.04

Note. For collective efficacy *df* = 238.82. SD = standard deviation

**Table 4 ijerph-19-13123-t004:** Multiple linear regression for pro-environmental behaviors (N = 402).

	B	SE	β	95% CI	*p*
Age	−0.01	0.01	−0.02	−0.01, 0.01	0.621
Years of education	−0.01	0.01	−0.05	−0.03, 0.01	0.241
Chronic disease	−0.03	0.07	−0.02	−0.16, 0.11	0.695
Climate change exposure	0.09	0.03	0.15	0.02, 0.16	0.008
Climate change risk appraisal	0.05	0.05	0.06	−0.04, 0.14	0.309
Environmental self-efficacy	0.24	0.04	0.29	0.16, 0.32	<0.001
Collective efficacy	0.07	0.04	0.11	0.01, 0.15	0.041

Adj. *R*^2^ = 0.17, *F* (7, 394) = 12.88, *p* < 0.001.

## Data Availability

The data that support the findings of this study are available from the authors upon reasonable request.

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
