# Peer review of "Factors Associated with Pro-Environmental Behaviors in Israel: A Comparison between Participants with and without a Chronic Disease"

_ijerph, 2022, doi:10.3390/ijerph192013123_

Round 1

Reviewer 1 Report

This study examines factors contributing to pro-environmental behaviors. The authors pack a lot of information into the manuscript, but it loses focus on the chronic disease aspect that is initially presented as the primary topic. While potentially important and novel research, the paper is missing a take home point of why this research matters. You touch on it as a driver of larger process change in the introduction, but that seems to be the closest the paper comes to a why. The paper needs significant rearranging to make it more digestible and to create a narrative that explains the importance of this research. In general it is also fairly wordy making it harder to follow, sentences should be tightened up to avoid extraneous words/phrases.

Abstract:

Ln 12: Does “appraisal” mean perceptions? Appraisal isn’t a typical word used in the field so the definition is a bit ambiguous.

Ln 15: You mention “higher” but how much higher? There should be some quantifiable information provided e.g. “…was associated with three-fold higher…” or “…higher climate change appraisal (score 4 vs 3, p=0.05)…”

Introduction:

Ln 36: leave just as “i.e. heat-related illness” and removed the examples after “such as”

Ln 44: I would remove “their” and the “s” from death so it says “may even lead to death”

Ln 52: move the aims more towards the end of the introduction, it is distracting to have it before you define PEBs or their importance.

Ln 62:  what does e.g 14 mean? if a reference cite the First author and year not just the ref #. otherwise delete e.g. and have just the ref #.

Ln 67-125: Move the framework model and the measure descriptions to the methods and only include a brief summary stating that the factors are complex in the introduction.

Ln 126-138: Listing predictors etc should be in the methods not introduction. This summary paragraph should be more to the point to explain what you did.

Ln 140-149: I would put the hypotheses in the methods section saying you tested the following hypotheses and leave the end of the introduction as pithier summary of what you did and why it’s important

Methods:

: mention the timeframe for when the survey was given, particularly since you mention the importance of this time in the discussion.

Ln 159: Is 402 the number of people you sent it to or the number who replied? I want to see the response rate. Did you send the survey to all 130000 panelists?

Ln 162-167: remove discussion of sample size calculations

Table 1: The table and all associated results describing the population that responded should be in the results, not methods.

Measures: Define/describe each measure, not just listing number of Q’s asked and one example.

Ln 183: If you used their entire questionnaire/tool from these referenced studies it's fine to just cite but if you didn't use their whole tool you should specify which measures you used. You can include the survey Qs as a supplementary material.

Ln 187: are these Cronbach’s α from the literature, or something you calculated? if you calculated this it should be in the results section. If you calculated also mention in statistical analysis section that you did so.

Ln 221: “calculated for the study variables by having a chronic disease” is confusing, rephrase.

Ln 224: What do you mean by background variables? Demographics?

Ln 227: “ NFI, NNFI, CFI, and RMSEA” Are these spss packages? need to clarify/define as they are not standard abbreviations.

Ln 229: What do you mean “were allowed to correlate among themselves?”

Ln 231: Provide more details on what variables were assessed for mediation, and provide a reference for why you chose to use bootstrapping

Results:

In general this section is overly wordy. Be more concise. You shouldn’t have to repeat methods definitions or reiterate things eg. “with regards to PEB, PEB was higher in this group.” The second part can stand alone.

Ln 261-266: This is the first time you mention “r” and “cramers V” outcomes. They should be mentioned/explained in the methods.

Ln 266-269: wordy “Further, smoking was associated with environmental self-efficacy such that participants who smoked (or used to) had lower environmental self-efficacy (Mean = 2.50, SD = 0.74) than participants who did not smoke (Mean = 2.68, SD = 0.72) (t(400) = 2.37, p = .018, d = 0.25).” Could just be “Further, current/former smokers had lower environmental self-efficacy (Mean = 2.50, SD = 0.74) than those who never smoked (Mean = 2.68, SD = 0.72) (t(400) = 2.37, p = .018, d = 0.25).”

:Besides the mention in ln 292-296, the results do not mention anything to do with the chronic disease. You do not report differences in path analysis by chronic disease etc (or at least don’t make it clear you were comparing the two groups). If you want to have the paper claim to focus on chronic disease these analyses should be performed and reported. If the sample size it too small to stratify by chronic disease status, chronic disease should be reported as a factor contributing to PEB/self efficacy etc, but not be the main focus of the title/introduction/abstract.

Discussion:

The first paragraph should emphasize the primary results, not just reiterate the purpose of the study. Since the topic is chronic disease, that should be a primary take away. Otherwise I would remove chronic disease as the focus and just look at predictors of PEB score etc (including chronic disease as an explanatory variable)

Ln 322: “moderate” is not a descriptive word. Moderate compared to what?

Ln 341: Rephrase to negative effects of climate change (shouldn't have to have “(worse)”) it should be clear with additional explanation

Ln 370: If chronic disease is the primary focus it should be higher up in the discussion.

Limitations: Add to limitations that PEBs are self-reported, which can lead to bias

:The discussion feels disjointed and doesn’t tell a story to make sense of the data. The confirmation of hypotheses doesn’t need to be the structure driving the discussion. The discussion should: a. summarize the results; b. explain whether it adds to/confirms previous research; c. describe implications of the data for policy interventions future research etc;

Author Response

October 2022

Dear Reviewer,

Re: Revised version of the manuscript:

Factors associated with pro-environmental behaviors in Israel: A comparison between participants with and without a chronic disease

 We would like to thank you for your insightful comments. In revising our manuscript, we paid close attention to each comment and accepted most of the recommended changes.

Below, we provide our detailed responses (in blue font). Please note that we numbered the comments.

We believe that the comments and the excellent suggestions, aimed at guiding us through the revision process, have significantly improved our manuscript.

Best regards,

Dr. Shiri Shinan-Altman
Prof. Yaira Hamama-Raz

Comments and Suggestions for Authors

  1. This study examines factors contributing to pro-environmental behaviors. The authors pack a lot of information into the manuscript, but it loses focus on the chronic disease aspect that is initially presented as the primary topic. While potentially important and novel research, the paper is missing a take home point of why this research matters. You touch on it as a driver of larger process change in the introduction, but that seems to be the closest the paper comes to a why.
    In the Introduction, we elaborated on the rationale for conducting this study specifically among people with a chronic disease. In addition, we put more of a focus on the chronic disease aspects throughout the paper (pp. 2-3).
  2. The paper needs significant rearranging to make it more digestible and to create a narrative that explains the importance of this research. In general it is also fairly wordy making it harder to follow, sentences should be tightened up to avoid extraneous words/phrases.

We have made our best efforts both to rearrange the manuscript and to tighten up the writing.

Abstract:

  1. Ln 12: Does “appraisal” mean perceptions? Appraisal isn’t a typical word used in the field so the definition is a bit ambiguous.

Thank you for this comment and question. In fact, previous studies in the field of climate change threats have indeed used the term "appraisal" (and, pursuant to your question and our wish to be as clear as possible, we have decided to use the term “risk appraisal” in the current paper). For example: Haas, S., Gianoli, A., & Van Eerd, M. (2021). The roles of community resilience and risk appraisal in climate change adaptation: the experience of the Kannagi Nagar resettlement in Chennai. Environment and Urbanization33(2), 560–578. https://doi.org/10.1177/0956247821993391

  1. Ln 15: You mention “higher” but how much higher? There should be some quantifiable information provided e.g. “…was associated with three-fold higher…” or “…higher climate change appraisal (score 4 vs 3, p=0.05)…”

These results stem from the path analysis and represent beta values between continuous variables. These beta scores are the quantification of the relationships and were added in the Abstract (p. 1).

Introduction:

  1. Ln 36: leave just as “i.e. heat-related illness” and removed the examples after “such as”.

We removed the examples after “such as.” (p. 2 line 44).

 6. Ln 44: I would remove “their” and the “s” from death so it says “may even lead to death”

We removed “their” and the “s” from death (p. 2 line 52).

 7. Ln 52: move the aims more towards the end of the introduction, it is distracting to have it before you define PEBs or their importance.

In accordance with the reviewer’s comment, the aims of the study appear toward the end of the Introduction (p. 4).

 8. Ln 62:  what does e.g 14 mean? if a reference cite the First author and year not just the ref #. otherwise delete e.g. and have just the ref #.

We removed the e.g. (p. 2 line 80).

 9. Ln 67-125: Move the framework model and the measure descriptions to the methods and only include a brief summary stating that the factors are complex in the introduction.

We thank the reviewer for this comment. However, this is a conceptual description of the components of our theoretical model. This conceptual description does not include the methodological parts of measurement which are presented in the Methods section. Therefore, we prefer to leave the description of the model and its components in the Introduction.

  1. Ln 126-138: Listing predictors etc should be in the methods not introduction. This summary paragraph should be more to the point to explain what you did.

We removed the predictors and the summary paragraph is now more to the point in terms of what we did (p. 4).

  1. Ln 140-149: I would put the hypotheses in the methods section saying you tested the following hypotheses and leave the end of the introduction as pithier summary of what you did and why it’s important.

As suggested, we put the hypotheses in the Methods section (p. 4).

Methods:

  1. mention the timeframe for when the survey was given, particularly since you mention the importance of this time in the discussion.
    We added that the data were collected during the first two weeks of June, 2022 (p. 5).
  2. Ln 159: Is 402 the number of people you sent it to or the number who replied? I want to see the response rate. Did you send the survey to all 130000 panelists?

We clarified that out of all 130,000 panelists, 5,378 potential participants were invited to participate in the study via email. Of them, 402 (7.5%) people completed the study’s questionnaire and participated in the study (p. 4). We added this low response rate to the limitations section (p. 12, lines 520-521).

  1. Ln 162-167: remove discussion of sample size calculations
    We moved the discussion of sample size calculations to the statistical analyses section (p. 6).
  2. Table 1: The table and all associated results describing the population that responded should be in the results, not methods.
    We moved Table 1 and all the associated results describing the participants to the Results section (pp. 4-5).
  3. Measures: Define/describe each measure, not just listing number of Q’s asked and one example.
    We added additional descriptions for each measure (pp. 5-6).
  4. Ln 183: If you used their entire questionnaire/tool from these referenced studies it's fine to just cite but if you didn't use their whole tool you should specify which measures you used. You can include the survey Qs as a supplementary material.
    We specified for each measure whether we used the whole scale or only a subscale, and we also added more item examples (pp. 5-6). In addition, we added supplementary material with the survey Qs.
  5. Ln 187: are these Cronbach’s α from the literature, or something you calculated? if you calculated this it should be in the results section. If you calculated also mention in statistical analysis section that you did so.

The Cronbach alphas were calculated for this study. This information was added to the statistical analyses section (p. 7). The values of the alphas were moved into Table 2 (p. 8).

  1. Ln 221: “calculated for the study variables by having a chronic disease” is confusing, rephrase.

We rephrased the sentence. It now reads: “Differences in the study variables, between participants with and without a chronic disease, were calculated with a series of t-tests” (p. 7).

  1. Ln 224: What do you mean by background variables? Demographics?
    We replaced the term "background" variables with "sociodemographic" variables (p. 7 line 286).
  2. Ln 227: “ NFI, NNFI, CFI, and RMSEA” Are these spss packages? need to clarify/define as they are not standard abbreviations.

These are measures of model fit within the AMOS-SPSS package. They were clarified in the statistical analyses section (p. 7).

  1. Ln 229: What do you mean “were allowed to correlate among themselves?”

We clarified: “Correlations among the covariates were included in the model, as well as between the two mediators of environmental self-efficacy and collective efficacy.” (p. 7, lines 298-299).

  1. Ln 231: Provide more details on what variables were assessed for mediation, and provide a reference for why you chose to use bootstrapping

We added a description of the independent, mediating, and dependent variables in the statistical analyses section (p. 7 lines 295-299). The rationale for choosing bootstrapping and a reference were added at the end of the statistical analyses section (p. 7 lines 302-304).

Results:

  1. In general this section is overly wordy. Be more concise. You shouldn’t have to repeat methods definitions or reiterate things eg. “with regards to PEB, PEB was higher in this group.” The second part can stand alone.

We removed method definitions and repetitive sentences in the Results section (pp. 8-9).

  1. Ln 261-266: This is the first time you mention “r” and “cramers V” outcomes. They should be mentioned/explained in the methods.

We now mention “r” and “Cramer’s V” in the statistical analyses section (p. 7, line 287).

  1. Ln 266-269: wordy “Further, smoking was associated with environmental self-efficacy such that participants who smoked (or used to) had lower environmental self-efficacy (Mean = 2.50, SD = 0.74) than participants who did not smoke (Mean = 2.68, SD = 0.72) (t(400) = 2.37, p = .018, d = 0.25).” Could just be “Further, current/former smokers had lower environmental self-efficacy (Mean = 2.50, SD = 0.74) than those who never smoked (Mean = 2.68, SD = 0.72) (t(400) = 2.37, p = .018, d = 0.25).”

We changed the abovementioned sentence. In addition, in order to avoid stigmatization we did not use “smokers,” but maintained the phrase “participants who smoke.” (p. 9 lines 362-363).

  1. Besides the mention in ln 292-296, the results do not mention anything to do with the chronic disease. You do not report differences in path analysis by chronic disease etc (or at least don’t make it clear you were comparing the two groups). If you want to have the paper claim to focus on chronic disease these analyses should be performed and reported. If the sample size it too small to stratify by chronic disease status, chronic disease should be reported as a factor contributing to PEB/self efficacy etc, but not be the main focus of the title/introduction/abstract.

Chronic disease appears in the study’s model, and results pertaining to it appear in the Abstract (p. 1 lines 15-20). In addition, we gave additional emphasis to it at the beginning of the model description in the Results section (p. 10 section 4.5). Further, it was stated as the key variable in one of the two main indirect relationships described about the model (p. 10 lines 188-390).

Discussion:

  1. The first paragraph should emphasize the primary results, not just reiterate the purpose of the study. Since the topic is chronic disease, that should be a primary take away. Otherwise I would remove chronic disease as the focus and just look at predictors of PEB score etc (including chronic disease as an explanatory variable).

We added the primary results to the first paragraph, beginning with the differences we found between participants with and without a chronic disease (p. 10 line 412).

  1. Ln 322: “moderate” is not a descriptive word. Moderate compared to what?

We added “relative to the scale range.” (p. 12 line 500).

  1. Ln 341: Rephrase to negative effects of climate change (shouldn't have to have “(worse)”) it should be clear with additional explanation.

We rephrased this sentence and added an additional explanation (p. 11, lines 440-443).

  1. Ln 370: If chronic disease is the primary focus it should be higher up in the discussion.

The paragraph that focuses on chronic disease is now higher up in the Discussion (p. 10 line 412).

  1. Limitations: Add to limitations that PEBs are self-reported, which can lead to bias.

We added this limitation to the limitations section (p. 13 lines 527-528).

  1. The discussion feels disjointed and doesn’t tell a story to make sense of the data. The confirmation of hypotheses doesn’t need to be the structure driving the discussion. The discussion should: a. summarize the results; b. explain whether it adds to/confirms previous research; c. describe implications of the data for policy interventions future research etc;

We changed the structure of the Discussion in accordance with the reviewer’s comment. We would like to mention that suggestions for future studies appear in the limitations section (p. 13).

Thank you!

Reviewer 2 Report

I read the manuscript with interest. The authors examined differences in pro-environmental behavior by comparing two different populations: one composed of individuals with chronic diseases and one composed of individuals without chronic diseases. The statistical approach allowed them to highlight the most important variables. The topic addressed is original, but there are some aspects that should be reconsidered to improve the manuscript.
- Some parts of the text could be moved to more appropriate sections. For example, there are some methodological considerations in the introduction (lines 70-71 and 80-81) that could be moved to the Materials and Methods section. Similarly, there are anticipations of results in Materials and Methods (lines 169-178).
- The authors examined the parameters of interest using surveys with specific items. Were validated questionnaires used? It would also be interesting to read any questions asked, which could be uploaded as supplementary material.
- In the materials and methods, the authors do not place their research in a temporal context. Only in the discussions, it is mentioned that the research was conducted in June. But in what year? And how many months did it take? If it took place in the last two years, the pandemic cannot be overlooked COVID -19 These are essential details that need to be added. 
- Regarding chronic diseases, the authors mention only a few examples (lines 179-171) but do not elaborate on the nature of the diseases. Since only 101 participants had chronic diseases, it would be very interesting to know which ones and how many of them were. This information should be added.
- An important limitation of the study that should be highlighted is that the duration of the chronic diseases was not considered. Also, possible drug therapies or subjective suffering were not considered. The section on limitations of the study should therefore be implemented. 
- There are some inaccuracies in the references that should be corrected, such as the date of access to the websites

Author Response

October 2022

Dear Reviewer,

Re: Revised version of the manuscript:

Factors associated with pro-environmental behaviors in Israel: A comparison between participants with and without a chronic disease

We would like to thank you for your insightful comments. In revising our manuscript, we paid close attention to each comment and accepted most of the recommended changes.

Below, we provide our detailed responses (in blue font). 

We believe that the comments and the excellent suggestions, aimed at guiding us through the revision process, have significantly improved our manuscript.

Best regards,

Dr. Shiri Shinan-Altman
Prof. Yaira Hamama-Raz

I read the manuscript with interest. The authors examined differences in pro-environmental behavior by comparing two different populations: one composed of individuals with chronic diseases, and one composed of individuals without chronic diseases. The statistical approach allowed them to highlight the most important variables.
Thank you.

The topic addressed is original, but there are some aspects that should be reconsidered to improve the manuscript.
- Some parts of the text could be moved to more appropriate sections. For example, there are some methodological considerations in the introduction (lines 70-71 and 80-81) that could be moved to the Materials and Methods section. Similarly, there are anticipations of results in Materials and Methods (lines 169-178).
We thank the reviewer for his/her suggestions. Accordingly, we moved text from the Material and Methods section to the Results section (p. 7). However, the transactional model of stress and coping that is mentioned in the Introduction section is the theoretical framework for this research design and as such we did not move it to the Material and Methods section.

The authors examined the parameters of interest using surveys with specific items. Were validated questionnaires used? It would also be interesting to read any questions asked, which could be uploaded as supplementary material.
We mentioned that all the questionnaires were validated and were used in previous studies in the area of climate change pro-environmental behaviors. In addition, we provided more details about each questionnaire including item examples (pp.5-6). We also created a file with the research questions as supplementary material.

In the materials and methods, the authors do not place their research in a temporal context. Only in the discussions, it is mentioned that the research was conducted in June. But in what year? And how many months did it take? If it took place in the last two years, the pandemic cannot be overlooked COVID -19 These are essential details that need to be added. 
We added that data collection was conducted during the first two weeks of June 2022, when the COVID-19 outbreak was under control and normal routines were in place (p. 5).

Regarding chronic diseases, the authors mention only a few examples (lines 179-171) but do not elaborate on the nature of the diseases. Since only 101 participants had chronic diseases, it would be very interesting to know which ones and how many of them were. This information should be added.
We elaborated on the information regarding types of diseases and the number of patients with each of these diseases (p. 7).

An important limitation of the study that should be highlighted is that the duration of the chronic diseases was not considered. Also, possible drug therapies or subjective suffering were not considered. The section on limitations of the study should therefore be implemented. 

We added these important limitations to the limitations section (p.13).

There are some inaccuracies in the references that should be corrected, such as the date of access to the websites
We checked and corrected the references list.

 Thank you!

Round 2

Reviewer 1 Report

Primary comments have been addressed.

The response rate should be put into the results not methods. In methods state you invited the 5k individuals, in results say 402 responded (7.5% response rate). 

ln 259 you say "A sample of 402 Israelis was selected..." you did not select the 402, these were who responded. This should be clarified.

Author Response

October 2022

Dear Reviewer,

Re: Revised version of the manuscript:

Factors associated with pro-environmental behaviors in Israel: A comparison between participants with and without a chronic disease

We would like to thank you again for your insightful comments. In revising our manuscript, we paid close attention to each comment and accepted the recommended changes.

We hope that you find the revisions satisfactory in making the manuscript suitable for publication in ‘IJERPH.’

Best regards,

Dr. Shiri Shinan-Altman
Prof. Yaira Hamama-Raz

Comments:

  1. The response rate should be put into the results not methods. In methods state you invited the 5k individuals, in results say 402 responded (7.5% response rate).

Response: We moved the response rate into the results. We stated that out of the 5,378 panelists, 402 (7.5%) people completed the study’s questionnaire and participated in the study (p. 6)

  1. ln 259 you say "A sample of 402 Israelis was selected..." you did not select the 402, these were who responded. This should be clarified.

Response: We clarified the abovementioned sentence (p. 6).

Thank you!

Reviewer 2 Report

I thank the authors for making the proposed changes. I have no other suggestions for further improving the manuscript. 

Author Response

Thank you!